# ^15^N Stable Isotope Labeling PSTs in *Alexandrium minutum* for Application of PSTs as Biomarker

**DOI:** 10.3390/toxins11040211

**Published:** 2019-04-08

**Authors:** Wancui Xie, Min Li, Lin Song, Rui Zhang, Xiaoqun Hu, Chengzhu Liang, Xihong Yang

**Affiliations:** 1College of Marine Science and Biological Engineering, Qingdao University of Science & Technology, Qingdao 266042, China; xiewancui@163.com (W.X.); 2017170007@mails.qust.edu.cn (M.L.); lylinsong@hotmail.com (L.S.); 2018170002@mails.qust.edu.cn (X.H.); 2Key Laboratory for Biochemical Engineering of Shandong Province, Qingdao 266042, China; 3College of Food Science and Technology, Guangdong Ocean University, Zhanjiang 524088, China; zhangr1168@163.com; 4Shandong Entry-Exit Inspection and Quarantine Bureau, Qingdao 266000, China; liangcz@163.com

**Keywords:** *Alexandrium minutum*, dinoflagellate, paralytic shellfish toxins (PSTs), ^15^N stable isotope labeling, biomarker

## Abstract

The dinoflagellate *Alexandrium minutum* (*A. minutum*) which can produce paralytic shellfish toxins (PSTs) is often used as a model to study the migration, biotransformation, accumulation, and removal of PSTs. However, the mechanism is still unclear. To provide a new tool for related studies, we tried to label PSTs metabolically with ^15^N stable isotope to obtain ^15^N-PSTs instead of original ^14^N, which could be treated as biomarker on PSTs metabolism. We then cultured the *A. minutum* AGY-H46 which produces toxins GTX1-4 in f/2 medium of different ^15^N/P concentrations. The ^15^N-PSTs’ toxicity and toxin profile were detected. Meanwhile, the ^15^N labeling abundance and ^15^N atom number of ^15^N-PSTs were identified. The ^14^N of PSTs produced by *A. minutum* can be successfully replaced by ^15^N, and the f/2 medium of standard ^15^N/P concentration was the best choice in terms of the species’ growth, PST profile, ^15^N labeling result and experiment cost. After many (>15) generations, the ^15^N abundance in PSTs extract reached 82.36%, and the ^15^N atom number introduced into GTX1-4 might be 4–6. This paper innovatively provided the initial evidence that ^15^N isotope application of labeling PSTs in *A. minutum* is feasible. The ^15^N-PSTs as biomarker can be applied and provide further information on PSTs metabolism.

## 1. Introduction

Harmful algal blooms (HABs) occur frequently in coastal areas worldwide, causing public concerns. Firstly, HABs cause millions of dollars economic losses in the tourism and industry sectors [1]. Secondly, HABs break the balance of marine ecosystems as they can disrupt communities and food web structures [2,3]. Thirdly, phycotoxins produced by HABs may be transferred through the food cycle, thereby causing lethal and sublethal effects on humans [4,5]. Among all the toxins produced by HABs species, paralytic shellfish toxins (PSTs) produced by dinoflagellates are the most widespread and potent shellfish contaminating biotoxins [6]. They are accumulated by filter-feeding bivalve mollusks and some zoophagous mollusks [7]. One of the most widespread toxigenic microalgal taxa is the dinoflagellate *Alexandrium,* and studies have demonstrated that toxic *Alexandrium spp*. disrupt behavioral and physiological processes in marine filter feeders [8,9]. In recent years, *A. tamarense*, *A. minutum*, *A. ostenfeldii*, and others were used as carriers of PSTs to track the migration, biotransformation, accumulation, and removal of PSTs under experimental conditions [10,11,12]. However, the metabolism of PSTs in bivalves is still unclear. Thus, there is a need for a new powerful tool to conduct PST-related researches.

Isotopic tracer technique can be used to study the absorption, distribution, metabolism, and excretion of some substances, such as protein and lipid, or to establish organism trophic links in the ecosystem [13,14,15,16]. The stable isotope labeling (ICAT, ICPL, IDBEST, iTRAQ, TMT, IPTL, and SILAC)-based quantitative approaches are highly efficient for obtaining highly accurate quantification results and for building more extensive small molecule databases [14,17], especially in proteomics [18,19]. Some studies were conducted on microalgae due to their ability to fix stable isotopes photosynthetically into cells and offer various target products [20,21,22,23]. Artificially enriched stable isotopes of nitrogen can be fed to cells, either in the form of ^15^N-labeled amino acid or ^15^N-labeled inorganic salt. According to [24], the microalgae *Chlamydomonas reinhardtii* was used to produce ^15^N-labeled amino acids with a high isotopic enrichment and sixteen ^15^N-labelled amino acids could be obtained.

However, isotope-labeling toxin-producing algae have been rarely studied so far. In the present study, we metabolically labeled the *A. minutum* produced gonyautoxins1-4 (GTX1-4) that belonging to mono-sulfated subgroup of PSTs [25]. The culture was carried out with ^15^N-NaNO_3_ as nitrogen source to obtain the ^15^N-PSTs that could be used as biomarker and provide further information on PSTs metabolism in filter feeding bivalve.

## 2. Results

### 2.1. Batch Culture: ^15^N/P Influence in Algal Growth and Toxicity

Initial batch culture experiments allowed comparison of algae growth under different ^15^N/P conditions, as follows: Group A: 1.0 time of the f/2 medium standard ^15^N/P concentration; Group B: 1.5 times of the f/2 medium standard ^15^N/P concentration; Group C: 2.0 times of the f/2 medium standard ^15^N/P concentration; Group D: 2.5 times of the f/2 medium standard ^15^N/P concentration; and Group E: 3.0 times of the f/2 medium standard ^15^N/P concentration. The results showed that ^15^N/P concentration can affect algal growth (Figure 1a). In an optimal culture environment, the growth curve of algae cell can be roughly divided into 4 phases, as follows: lag phase 0–15 days; log phase 16–30 days; 31 days of the stable phase; and decay phase. In this study, the decay phase was not studied, because the focus was on the degree of ^15^N labeling of algal cellular toxin. The results were similar to those obtained in the report [26]. However, the lag phase of this experiment was much longer, and it may have been caused by partial mechanical damage to cells when algal cells were collected by centrifugation. After the growth of algal cells into the log phase, Group B cell density was higher than those of the other four groups (*p* < 0.05). At 30 days, the maximum value of Group B reached 3.820 × 10^4^ cells/mL, and there was no significant difference in Group A cell density (3.342 × 10^4^ cells/mL) (*p* > 0.05). However, the biomass of Groups A and B biomass was much higher than that of the other three groups (*p* < 0.01). Lower biomass at high nutrient concentration could be attributed to the inhabitation of photosystem II’s photosynthetic capacity at high nutrient levels [27].

^15^N/P conditions also can affect toxicity of algae cells (Figure 1b). The highest toxin levels were determined at day 25 in the log phase in all groups, the same as [28], not early- or post- stationary growth phase mentioned in [29,30]. Concerning the factors influencing toxicity, the toxin content per cell in batch culture was not only related to the cell growth stage, but was also affected by intracellular nutrient salts (e.g., nitrogen, phosphorus, and carbon dioxide), thereby reflecting the balance between synthesis and leakage of toxins (e.g., catabolism and cell division) [31].

### 2.2. Generation to Generation Culture: ^15^N/P Influence in Algal Growth and Toxicity

The algal cells of Groups A and B were chosen via generation to generation culture (three generations) because of better growth. The effects of ^15^N/P concentration on cell growth and toxicity are displayed in Figure 2. In the first generation, Group B algae cell growth was better than that of Group A; the same results were obtained in batch culture experiments. In the second and third generations, the cell density of Group B was less than that of Group A, especially in the third generation. Interestingly, there was no significant difference (*p* < 0.05) in algal cytotoxicity between the two groups in all three generations. The experimental results showed the algae cells cultured in a nutrient solution with a higher ^15^N/P than f/2 medium standard ^15^N/P concentration gradually deteriorated, indicating that f/2 medium standard ^15^N/P concentration was more suitable for domesticating high abundance ^15^N-PST *A. minutum*.

### 2.3. ^15^N/P Effect on Algae PSTs Profile

The experiment was carried out using 1.0 and 1.5 times of the f/2 medium standard ^15^N/P concentration compared with 1.0 and 1.5 times of the f/2 medium standard ^15^N/P concentration. The experimental *A. minutum* only produced GTX1-4; the major components are GTX-2,-3, which accounted for about 74% of the total toxin (Figure 3). Changing the standard N/P concentration influenced the profile of PSTs, and no significant difference was found between 1.0 time of the f/2 medium standard ^15^N/P and ^14^N/P concentration, indicating that the application of ^15^N isotope labeling can feasibly be used to study PSTs production and metabolism.

### 2.4. ^15^N Labeling Abundance Change of ^15^N-PSTs

The ^15^N labeling abundance was calculated according to the following formula:^15^N atom% = 1/(2I_28_/I_29_ + 1) × 100 (^15^N atom% < 10%);^15^N atom% = 2/(I_29_/I_30_ + 2) × 100 (^15^N atom% > 10%)(1)

In the algal culture environment, ^15^N-NaNO_3_ of f/2 medium is not the only nitrogen source because of the inorganic ^14^N existing in natural seawater. *A. minutum* cells prefer absorbing light elements (^14^N) and rejecting heavy elements (^15^N), resulting in nitrogen stable isotope fractionation. With increasing cell density and size, less and less ^14^N can be utilized. Thus, more ^15^N can enter cells, and nitrogen stable isotopes fractionation weakens. Two culture methods and gas isotope mass spectrometer were used to determine the relationship between ^15^N-labeling abundance and culture time, as shown in Table 1. During the batch culture, ^15^N abundance was positively related to culture time and reached the highest value at day 30. There are significant differences (*p* < 0.01) in terms of ^15^N abundance among lag, log, and stable phases. After many generations, ^15^N abundance reached to 82.36 atom%, but this percentage was still far below the abundance of the labeling material ^15^N-NaNO_3_ (δ^15^N = 99.14%), not reaching our expectation (>90%).

### 2.5. The Efficiency of ^15^N-PSTs Separation and Purification

^15^N-PSTs extracts were separated and purified by column chromatography on the Bio-Gel P-2 and the weak cation exchanger Bio-Rex 70. In the process of column chromatography on the Bio-Gel P-2, fluorescence detection and UV absorbance detection were carried out (Figure 4a). The UV absorption peak did not coincide with the fluorescence absorption peak. Thus, the UV absorption signal had no relationship with the toxin component. As shown by the fluorescence absorption peak, Bio-Gel P-2 effectively separated ^15^N-PSTs with impurities in crude extracts, such as proteins and pigments. The liquid of fluorescence absorption peak was collected, freeze-dried (10 mg), and redissolved with 0.05 M Tri-HCl (2 mL). The redissolved sample (1 mL) was used for purification by column chromatography on weak cation exchanger Bio-Rex 70 at gradient elution condition and was separated (Figure 4b). After analysis, the Peak Ⅰ was determined to be the isomer mixture of GTX1/4, and Peak Ⅱ was the isomer mixture of GTX2/3.

### 2.6. ^15^N Atom Number Identification of ^15^N-PSTs

As demonstrated previously [32], GTX1-4 (Figure 5) can be ionized in ESI positive ion mode, thereby giving abundant fragment ions (Table 2). When ^14^N of PSTs is replaced by ^15^N, [M+H]^+^ and fragment ions will change with the number of ^15^N atoms, so that primary mass spectrum can be used to determine the ^15^N atom number of ^15^N-PSTs. After the addition of artificial nitrogen in *A. minutum* culture, ^15^N was successfully introduced to PSTs, and characteristic fragment ions were produced. There was no [M+H]^+^ peak in the mass spectrum as the result of high capillary voltage in this experiment (Figure 6). GTX1 produced three characteristic fragment ions (m/z = 334.2966; 318.3007; 319.3054) with the loss of -SO_3_ or -SO_3_-H_2_O. GTX4 generated three characteristic fragment ions (m/z = 318.3007; 319.3054; 338.3418) with the loss of -SO_3_-H_2_O or -SO_3_ and had two fragment ions (m/z = 318.3007; 319.3054) similar to those of GTX1. There were four characteristic fragment ions (m/z = 322.1197; 321.1252; 305.1583; 317.0497) in GTX2 with the loss of -SO_3_ or -SO_3_-H_2_O, whereas two characteristic fragment ions (m/z = 302.3054; 303.3095) were obtained with the loss of -SO_3_-H_2_O. Theoretically, the m/z of fragment ions can increase (+1,+2,+3,+4,+5,+6,+7) corresponding with the number (1,2,3,4,5,6,7) of introduced ^15^N atoms in PSTs. Based on these data, 2~6 ^15^N atoms can be introduced into the ^15^N-PSTs molecule, and 4~6 ^15^N atoms is most possible.

## 3. Discussion

Numerous studies have focused on the bioaccumulation and biotransformation of PSTs using *Alexandrium* strains, such as *A. minutum* in marine organisms [33]. Stable isotopes have been often used to study lipid synthesis, proteomics and ecosystem [13,15,34] and rarely applied to toxin-producing algae [35]. This study intends to use the biosynthetic process to replace the nitrogen atom of *A. minutum* with a stable isotope ^15^N to form a tracer-enabled *A. minutum* for PST synthesis and metabolism.

### 3.1. Effect ^15^N/P of on A. minutum Culture

Growth and toxin production of toxic dinoflagellates vary with nutrients supply. High nutrient cultures can inhibit the photosynthetic capacity of photosystem II, which related to algae growth [27]. The N:P ratio can be used as an index for the nutritional status and physiological behavior of phytoplanktons [36]. Kinds of nitrogen sources and N:P supply ratio can affect the physiological responses of a tropical Pacific strain of *A. minutum*, the cellular toxin quota (Qt) was higher in P-depleted, nitrate-grown cultures [37]. To assure the feasibility of isotope ^15^N, some experiments were carried out, including the comparison of different ^15^N:P ratio. In batch culture, five ^15^N:P ratios were compared, and the growth of *A. minutum* under 1.0 and 1.5 times of the f/2 medium standard ^15^N/P concentration was better than others (Figure 1a). Similar previous findings reported that cell densities and growth rates of *A. minutum* were severely suppressed under high N/P ratios (>100) in both N-NO_3_ and N-NH_4_ treatments [38]. Some studies showed that the incorporation of ^15^N-labeled salt did not affect the growth of green alga *Chlamydomonas reinhardtii* [24,39], and our study achieved the same result with *A. minutum*. The toxin profile of this *A. minutum* strain is relatively stable and predominantly constitutes GTX1-4 (Figure 3), the same as four strains of *A. minutum* collected from southern Taiwan [40], even under different N:P supply ratios. The highest algae toxicity per cell of all five groups was observed at day 25, and cellular toxin quota of the exponential growth phase was higher than that of the stable phase, even though the total number of stable cells is highest during the entire growth process (Figure 1b). The explanation could be that there was a negative correlation between algae toxicity per cell and cell density. In other words, the cell size was smaller when cell density was higher; thus, toxicity per cell of stable phase was lower. As previously reported, changes of nutrient availability with time in batch culture caused growth stage variability in toxin content, which peaked during mid-exponential growth [31]. Total toxicity, toxicity per cell, and the number of and relative proportion of toxin analogs changed in relation to the ^15^N:P ratio. The f/2 medium standard ^15^N/P concentration at 1.0 time was a better choice to label PSTs with ^15^N regardless of cultivation method, i.e., batch culture or generation to generation culture, for *A. minutum* growth, PST production and profile, and experimental cost.

### 3.2. The Replacement of Stable Isotope ^15^N

The successful production PSTs of labeled substances from *A. minutum* was detected by MAT-271 Gas isotope mass spectrometer to determine ^15^N abundance, Analysis by HPLC-MS was performed to identify ^15^N atom number. In batch culture, δ^15^N of two groups’ PSTs had a significant difference (*p* < 0.01) in different growth stages (lag, log, and stable phases) (Table 2). Combining ^15^N labeling abundance with mass spectrum results, it can be presumed that artificial ^15^N addition can bring 2~6 ^15^N atoms into the ^15^N-PSTs molecule, and the 4~6 ^15^N atoms’ replacement becomes most possible. Recently, ^15^N stable-isotope-labeling was applied to the toxic dinoflagellate *Alexandrium catenella* and the relationship between the order of ^15^N incorporation % values of the labeled populations and the proposed biosynthetic route was established [35]. Relative abundance % of m+6 and m+7 isotopomers of PSTs were the highest in *A. catenella* after a two month passage in ^15^N-NaNO_3_ medium in [35], which is similar to our conclusion. Nitrogen (N) isotopic compositions of PSTs in *A. minutum* cells reflect the isotopic fractionations associated with diverse biochemical reactions. Based on PSTs being a secondary metabolite, a small part of the supplied nitrogen was assimilated into PSTs, and most of the nitrogen may participate in the synthesis of other nitrogen compounds. Some findings on high abundance in biomass (not extracted) have been reported; a process for the cost-effective production of ^13^C/^15^N-labelled biomass of microalgae on a commercial scale is presented, and 97.8% of the supplied nitrogen is assimilated into the biomass [23]. However, in the present paper, ^15^N-labeling abundance of the ^15^N-PSTs extract was 82.36%, lower than the abundance of whole cell in existing research [23]. The occurrence of this situation can be explained by the following reasons. Firstly, the time of each generation culture is not sufficiently long enough. Secondly, the total ^15^N abundance in the crude extract of the toxin cannot fully represent ^15^N in the pure toxin, because extract impurities (e.g., protein and pigment) can cause interference. Thirdly, trace amounts of nitrogen in natural seawater and the culture solution reagent may affect ^15^N-labeled PSTs.

## 4. Conclusions

This paper provides the initial evidence that ^15^N isotope is feasible to label PSTs in *A. minutum* and worthy of being a powerful tool to conduct PST-related researches. In our study, ^15^N abundance, PSTs content and profile were detected and the ^15^N atom number introduced into GTX1-4 should be 4–6. However, it is a pity that the precursor and the biosynthetic intermediates of PSTs in *A. minutum* were not analyzed. Further study is needed to apply isotope-labeling on both toxin-producing algae and vector mollusk species so that we can better elucidate the mechanism of PSTs biosynthesis and metabolism.

## 5. Materials and Methods

### 5.1. Chemicals and Analytical Standards

^15^N-NaNO_3_ (δ^15^N = 99.14%) was obtained from SRICI (Shanghai Research Institute of Chemical Industry CO., LTD., Shanghai, China). Bio-gel P-2 (400 mesh), Bio-Rex 70 (400 mesh) were obtained from BIO-RAD (Hercules, CA, USA). Certified reference materials for PSTs, including gonyautoxin 1/4 (GTX 1/4), gonyautoxin 2/3 (GTX 2/3), were purchased from the National Research Council, Institute for Marine Bioscience (Halifax, Canada). Analytical grade solvents were used for extraction purposes while LC grade solvents were used for HPLC-FLD applications.

### 5.2. Algal Culture

The PSTs-producing dinoflagellate *A. minutum* (strain AGY-H46, purchased from Leadingtec, Shanghai, China) was cultivated in thermo regulated rooms (25 ± 1 °C) with filtered (0.45 μm, Jinjing Ltd., China) and sterilized (121 °C, 20 min) seawater before enrichment with f/2 medium amendments (Table 3). The light intensity was set at 3000–4000 lux with a dark:light cycle of 14:10 h. The seawater was obtained from Donghai Island waters (Zhanjiang, China). Algal cell densities were determined by optical microscope and cells were collected at particular time for ^15^N abundance analysis and PSTs detection.

### 5.3. ^15^N-PSTs Extraction

An aliquot (60 mL) of the algal fluid was centrifuged at 6000 r/min under 4 °C for 10 min, and the supernatant was carefully discarded. The sedimentary cells were resuspended with 0.05 M acetic acid and then broken using ultrasonic processor in an ice bath for 10 min (power 80%, working 3 s, gap 3 s) until there was no whole cell. The combined liquid was centrifuged at 12000 r/min under 4 °C for 10 min, then the supernatant was filtered (0.22 μm, Jinjing Ltd., China) and stored under −20 °C for purification.

### 5.4. ^15^N-PSTs Separation and Purification

Separation and purification were carried on by reference to published papers [41,42]. The PSTs extract was adsorbed to a column of Bio-Gel P-2 equilibrated with water. Furthermore, the column was first washed with a sufficient volume of water and eluted with 0.1 mM acetic acid at a flow rate of 0.5 mL/min. After separation, toxin mixture was purified by ion exchange chromatography using a column of Bio-Rex 70 equilibrated with water and gradient eluted with acetic acid (acetic acid concentration was as follows: 0, 0.05, 0.055 and 0.060 mM). The fraction was collected every 12 min and then analyzed by FFA to assure purification efficiency.

### 5.5. ^15^N Abundance Analysis of ^15^N-PSTs Extracts

^15^N abundance was analyzed by MAT-271 Gas isotope mass spectrometer (Finnigan, Santa Clara Valley, CA, USA). Operating conditions were set to high voltage (10 kV), emission current (0.040 mA), electronic energy (100 eV) and well voltage (134 eV). ^15^N-PSTs extraction after freeze-drying was converted to gas by micro-high-heat combustion method, then entered into gas isotope mass spectrometer sample introduction system via sample adapter. Mass spectrometer vacuum was less than 2 × 10^−5^ Pa. After making the necessary calibration settings for the instrument, the instrument background measurement was performed. The sample gas was introduced into the sample storage system at a pressure of 5–10 Pa. The measurement process was automatically performed by computer instructions, and the signal strength (in mV) of mass number 28, 29, 30 was output as I_28_, I_29_ and I_30_.

### 5.6. PSTs Toxicity Test by the Mouse Bioassay

The mouse bioassay (MBA) was a referenced method from [43]. 10 healthy male mice were injected intraperitoneally with 1 mL aliquot of ^15^N-PSTs extracts and observed to quantify the toxin according to the time of death. The toxicity was expressed in mouse units (MU), 1 MU representing the average toxin amount to kill a mouse weighing 20 g within 15 min.

### 5.7. ^15^N-PSTs Detection by the Fast Fluorimetric Assay (FFA)

The fast fluorimetric assay (FFA) was performed by fluorospectrophotometer (HITACHI, Japan) at a 333 nm excitation wavelength, 10 nm excitation slit, emission wavelength 390 nm and 20 nm emission slit. The method was from [44] and got some modification in this paper. A portion (0.5 mL) of each extract or blank solution (0.05 M acetic acid), respectively, was mixed with 2 mL of oxidation solution (50 mM dipotassium phosphate with 10 mM periodic acid) and incubated for 15 min at 50 °C. After incubation, the reaction mixture was neutralized with 2.5 mL of 1 M acetic acid and transferred to a cuvette for detection by the fast fluorimetric assay (FFA). Relative fluorescence units (RFU) was recorded and this experiment was conducted to get an overview of the fluorescence of oxidized samples, but not for accurate quantification.

### 5.8. ^15^N-PSTs Determination by HPLC-FLD

HPLC-FLD analysis was carried out using Agilent 1100 (Agilent, Santa Clara, CA, USA) coupled to PCX5200 (Pickering Laboratories Company) post-column reactor and Agilent G1321A detector (λ_ex_ = 330 nm, λ_em_ = 390 nm). Chromatographic separation of compounds was achieved on a reversed-phase C8 column (150 mm × 4.6 mm i.d.; 5 μm, GL Sciences, Tokyo, Japan). The system was managed by an Agilent Chem. Station 2.1 workstation. The PST extract was subjected to HPLC-FLD after ultrafiltration (10,000 Da ultrafiltration centrifuge tube, 12,000 r/min for 10 min at 4 °C). Analysis method was performed according to [45]. The oxidant solution was 50 mM potassium hydrogen phosphate containing 7 mM periodic acid. The acidifying agent was 0.5 M acetic acid. The flow rates were 0.8 mL/min for the elution solution and 0.4 mL/min for the oxidant and acidifying agent.

### 5.9. ^15^N-PSTs Quantification by HPLC-MS

HPLC-MS analysis was carried out on Agilent 1100 (Agilent, USA) coupled to Waters FIN_C2-XS QTof mass spectrometer (Waters, Milford, MA, USA) with an electrospray ionization interface. PSTs were separated on a TSKgel Amide-80 HILIC column (250 mm × 2 mm i.d.; 5 μm, Tosoh Bioscience, LLC, Montgomeryville, PA, USA). The PSTs extract after purification was subjected to HPLC-MS after ultrafiltration (10000 Da ultrafiltration centrifuge tube, 12000 r/min for 10 min at 4 °C). A binary mobile phase included “solvent A” and “solvent B”, in which “solvent A” was water containing 2.0 mM ammonium formate containing 0.1% (*v*/*v*) formic acid and “solvent B” was acetonitrile containing 0.1% (*v*/*v*) formic acid. Parameters of mass spectrometer were multiple reaction monitoring (MRM) mode, positive polarity of ESI, capillary voltage (3.2 kV), ion source temperature (150 °C), desolvation temperature (400 °C), cone gas flow (50 L/h) and desolvation gas flow (700 L/h).

## Figures and Tables

**Figure 1 toxins-11-00211-f001:**
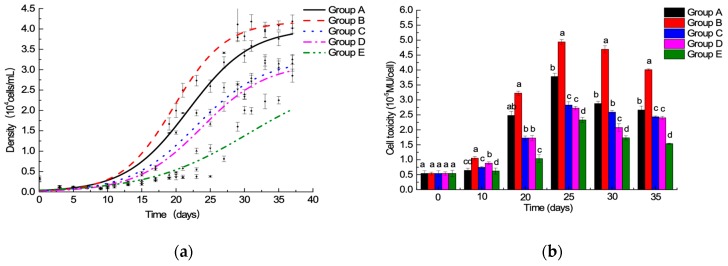
The effects of ^15^N/P concentration on cell growth (**a**) and toxicity (**b**) in batch culture. (——, −−−−, ∙∙∙∙∙∙∙∙, −∙−·−∙ and −∙∙−∙∙−∙∙: Growth/Sigmoidal. Letters indicate significant differences between conditions).

**Figure 2 toxins-11-00211-f002:**
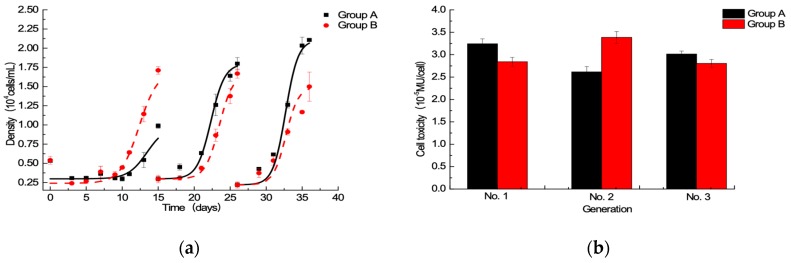
Effects of ^15^N/P concentration on cell growth (**a**) and cell toxicity (**b**) in generation to generation culture.

**Figure 3 toxins-11-00211-f003:**
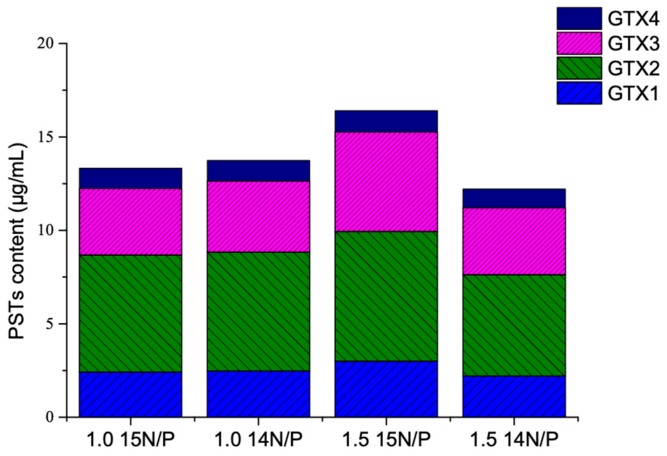
The effects of ^15^N/P concentration on *A. minutum* PST profile.

**Figure 4 toxins-11-00211-f004:**
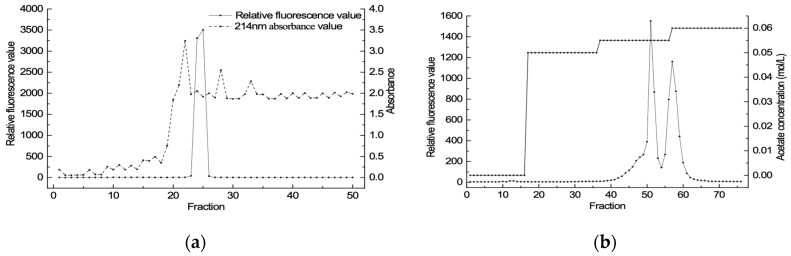
Gonyautoxin1-4 purified by column chromatography on the Bio-Gel P-2 (**a**) and the weak cation exchanger BioRex 70 (**b**).

**Figure 5 toxins-11-00211-f005:**
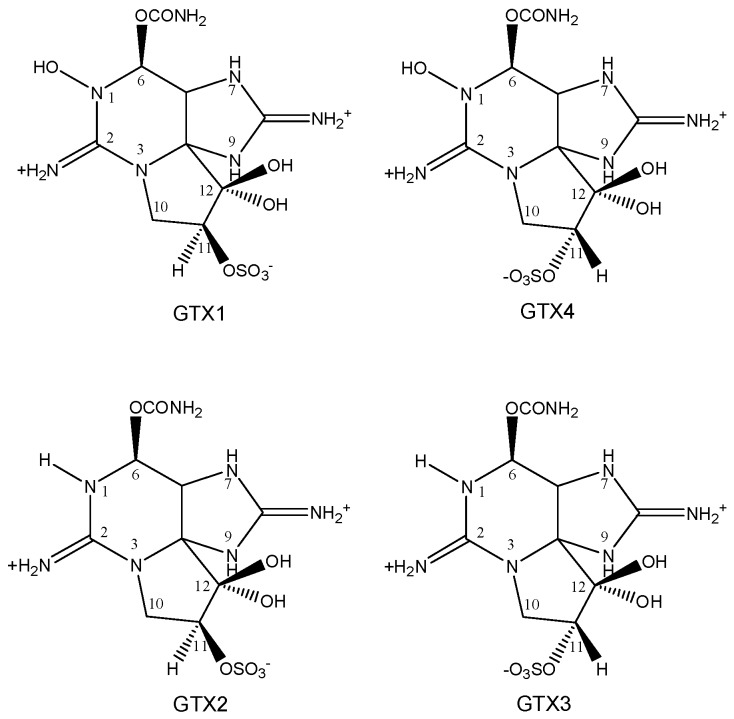
The molecular structural formula of GTX1-4.

**Figure 6 toxins-11-00211-f006:**
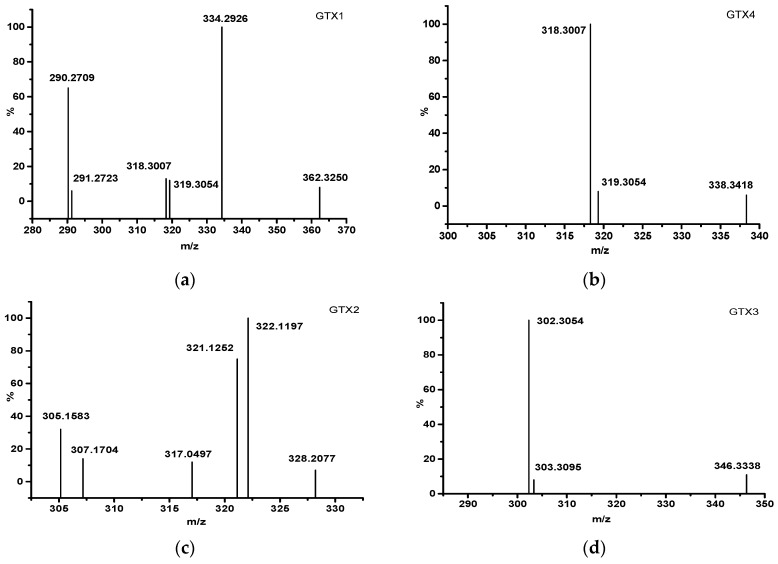
Mass spectrum of ^15^N-GTX1 (**a**), ^15^N-GTX4 (**b**), ^15^N-GTX2 (**c**) and ^15^N-GTX3 (**d**).

**Table 1 toxins-11-00211-t001:** ^15^N labeling abundance change of PSTs along with the change of culture time in different cultivation methods.

Culture Method		1.0 Time of the f/2 Medium Standard ^15^N/P Concentration	1.5 Times of the f/2 Medium Standard ^15^N/P Concentration
		PSTs ^15^N Abundance (Atom%)	
Batch culture/d	0	0.47	0.47
	20	26.26	26.43
	30	57.16	70.26
Generation to generation culture/generation	1	37.60	-
	2	58.32	-
	3	62.46	-
No. n generation	-	82.36	

**Table 2 toxins-11-00211-t002:** Mass spectral data for gonyautoxins (GTX1-4).

Toxin	Molecular Formula	[M+H]^+^	Fragment Ion	Loss of	Fragment Ion after Labeling
GTX1	C_10_H_17_N_7_O_9_S	412	332; 314	-SO_3_; -SO_3_-H_2_O	334.3; 318.3; 319.3
GTX4	C_10_H_17_N_7_O_9_S	412	332; 314;253	-SO_3_; -SO_3_-H_2_O; -SO_3_-H_2_O-NH_3_-CO_2_	318.3; 319.3; 338.3
GTX2	C_10_H_17_N_7_O_8_S	396	316; 298	-SO_3_; -SO_3_-H_2_O;	322.1; 321.1; 305.1
GTX3	C_10_H_17_N_7_O_8_S	396	316; 298; 220	-SO_3_; -SO_3_-H_2_O; -SO_3_-2H_2_O-NH_3_-NHCO	302.3; 303.3

**Table 3 toxins-11-00211-t003:** f/2 medium amendments.

Reagent		Working Solution (mg/L)	Stock Solution (g/L)
A:	NaNO_3_	75	75
B:	NaH_2_PO_4_·H_2_O	5	5
C:	Na_2_SiO_3_·9H_2_O	20	20
D:	Na_2_EDTA	4.36	4.36
E:	FeCl_3_·6H_2_O	3.16	3.16
F:	CuSO_4_·5H_2_O	0.01	0.01
	ZnSO_4_·7H_2_O	0.023	0.023
	CoCL_2_·6H_2_O	0.012	0.012
	MnCL_2_·4H_2_O	0.18	0.18
	Na_2_MoO_4_·2H_2_O	0.07	0.07
G:	Vitamin B1	0.1	0.01
	Vitamin B12	0.5 × 10^−3^	0.5 × 10^−4^
	Vitamin H	0.5 × 10^−3^	0.5 × 10^−4^

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
