# Peer review of "15N Stable Isotope Labeling PSTs in Alexandrium minutum for Application of PSTs as Biomarker"

_toxins, 2019, doi:10.3390/toxins11040211_

Round 1
Reviewer 1 Report
This is an interesting paper that will be useful to those hoping to prepare stable, isotope labeled PSTs for various studies.
There is some revision required:
Regarding the experimental methods section, reference to the following citation was used to explain what HPLC-FD method was used.
The experimental section needs revision and so do some literature citatons.
Instead of clearly citing which HPLC-FD method was used the authors took a confusing approach by citing the reference below in the following text on page 10, lines 306-307:
"Analysis 306 method was performed according to [36]"
Reference 36:
Costa, P.R.; Baugh, K.A.; Wright, B.; Ralonde, R.; Nance, S.L.; Tatarenkova, N.; Etheridge, S.M.; Lefebvre, 427 K.A.Comparative determination of paralytic shellfish toxins (PSTs) using five different toxin detection 428 methods in shellfish species collected in the Aleutian Islands, Alaska. Toxicon : official journal of the 429 International Society on Toxinology 2009, 54, 313-320, doi:10.1016/j.toxicon.2009.04.023. 430
431
The above reference is not appropriate for the precolumn oxidation HPLC method cited in the reference above. It is, as the title states, a comparison of results from several different methods.Instead they should dirctly cite the following if they did indeed use the AOAC pre-column HPLC method.
J.F. Lawrence, B. Niedzwiadek, C. Menard
Quantitative determination of paralytic shellfish poisoning toxins in shellfish using prechromatographic oxidation and liquid chromatography with fluorescence detection: collaborative study. J. AOAC Int., 88 (2005), pp. 1714-1732.
Unfortunately even more confusion is introduced in the experimental section because the HPLC equipment described (page 10, lines 301-309) is that which would be used for POST-COLUMN detection rather than the precolumn detection method cited in the Costa et al 2009 paper above. The authors must list proper references and not introduce confusion with literature that does not match the experimental conditions used. It would seem that the authors actually used a postcolumn-HPLC method since they list separately list results for individual toxins GTX2, GTX3 and GTX1, GTX4 rather than epimer pairs (like GTX2/GTX3 and GTX1/GTX4) which cannot be determined by the above precolumn method of Lawrence et al.
This paper should not be published until the above problems are clarified.
Author Response
Point 1: Regarding the experimental methods section, reference to the following citation was used to explain what HPLC-FD method was used. The experimental section needs revision and so do some literature citatons. Instead of clearly citing which HPLC-FD method was used the authors took a confusing approach by citing the reference below in the following text on page 10, lines 306-307:
"Analysis 306 method was performed according to [36]"
Reference 36:
Costa, P.R.; Baugh, K.A.; Wright, B.; Ralonde, R.; Nance, S.L.; Tatarenkova, N.; Etheridge, S.M.; Lefebvre, 427 K.A.Comparative determination of paralytic shellfish toxins (PSTs) using five different toxin detection 428 methods in shellfish species collected in the Aleutian Islands, Alaska. Toxicon : official journal of the 429 International Society on Toxinology 2009, 54, 313-320, doi:10.1016/j.toxicon.2009.04.023. 430
Response 1: Thank you very much for the comments concerning our manuscript and we are very sorry for our negligence. We read the reference [36] carefully and found it is not appropriate here. The method we used was post-column HPLC and reference [36] before has been deleted and changed to the reference below:
Van, d.R.J.; Gibbs, R.S.; Muggah, P.M.; Rourke, W.A.; Macneil, J.D.; Quilliam, M.A. Liquid chromatography post-column oxidation (PCOX) method for the determination of paralytic shellfish toxins in mussels, clams, oysters, and scallops: collaborative study. Journal of AOAC International 2011, 94, -, doi:AOAC International.(page 14, lines 469-472)

Reviewer 2 Report
1. Lines 75-77: ".....inhibited after exceeding a certain range." - please give some details
2. The references of discussions are relevant and comparable, but lack of some literature to support major results. Lines 169-171.
3. The conclusions described the data results or restated the data. Please re-write it.
Author Response
Point 1: Lines 75-77: ".....inhibited after exceeding a certain range." - please give some details.
Response 1: First of all, thanks for your useful comments. The data showed lower biomass at 3 higher nutrient concentration. The reason may be the photosynthetic capacity of photosystem II(PSII) of A. minutum was inhibited in high nutrient cultures. As for this, we add a reference "Effects of salinity and nutrients on the growth and chlorophyll fluorescence of Caulerpa lentillifera" (page 13, lines 417-419), which reported that the inhibition of PSII occurred when NO3-N concentrations were greater than 1.0 mmol/L and when PO4-P concentrations were at 0.4 mmol/L. But the paper added is about Caulerpa lentillifera not dinoflagellate we used, so there may be a little difference between two different strain. The accurate inhibition concentration of dinoflagellate A. minutum we used still need to be study in the future.
Point 2: The references of discussions are relevant and comparable, but lack of some literature to support major results. Lines 169-171.
Response 2: According to your comment, we have made some changes in the text. There are some papers referenced to support the results about A. minutum growth and PSTs profile such as lower biomass at high nutrient concentration and 15N-labeled salt did not affect the algae growth (page 7, lines 183-195). But, we have to admitted the lack of reference about the results of 15N isotope-labeling PSTs. Because the related research is very limited. Fortunately, a Japanese research group published their paper Metabolomic study of saxitoxin analogues and biosynthetic intermediates in dinoflagellates using (15)N-labelled sodium nitrate as a nitrogen source on Scientific reports (page 13, lines 410-413) recently(2019.03.05). We added this paper and compared to our results in revised manuscript (page 8, lines 222-226).
Point 3: The conclusions described the data results or restated the data. Please re-write it.
Response 3: We have re-written conclusion section as your suggestion. The revised conclusion is as follows.
This paper provides initial evidence that 15N isotope application of labeling PSTs in A. minutum is feasible and worthy of being a powerful tool to conduct PST-related research. In our study, 15N abundance, PSTs content and profile were detected. But it is a little pity that the precursor, and the biosynthetic intermediates of PSTs in A. minutum were not been analyzed. Further study is needed to apply isotope-labeling on both toxin-producing algae and vector mollusk species so that we can elucidate the mechanism of PSTs biosynthesis and metabolism better.(page 8, lines 244-249)

Reviewer 3 Report
Page 3, lines 79-85. How did you know it is the toxin that you are measuring based on cell density? Did you perform cytotoxicity assay on it?
Make your paper more organized. I cannot get the main point directly. Write simply.
Author Response
Point 1: Page 3, lines 79-85. How did you know it is the toxin that you are measuring based on cell density? Did you perform cytotoxicity assay on it?
Response 1: Thank you for pointing this out. When we detected PSTs toxicity, we also determined algal cell densities by optical microscope (mentioned in 5.2. Algal culture,page 9, lines 275-277). Toxin content per cell can be calculated by total toxicity and cell densities. We didn’t perform cytotoxicity assay. To avoid ambiguity, “cytotoxicity” has been replaced by “toxicity of algae cells ” or “toxin content per cell”. Besides, 3 references were added this time for better comparison. In our experiments, the highest cellular toxin levels were determined at the log phase (Day 25), like the reference Effects of nutrient limitation on toxin production and composition in the marine dinoflagellate Protogonyaulax tamarensis (page 13, lines 420-422). But in most existing research, the highest Qt (cellular toxin content ) often came out in stationary growth phase. For example, early-stationary growth phase in Variability in Toxicity of the Dinoflagellate Alexandrium Tamarense Isolated from Hiroshima Bay, Western Japan, as a Reflection of Changing Environmental Conditions (page 13, lines 423-426) and post-stationary growth phase in Influence of environmental and nutritional factors on growth, toxicity, and toxin profile of dinoflagellate Alexandrium minutum (page 13, lines 427-429) and Growth and toxin production in batch cultures of a marine dinoflagellate Alexandrium tamarense HK9301 isolated from the South China Sea (page 13, lines 430-432). The different pattern of toxin production which produced the highest Qt in log phase should be caused by the strain we used.
Point 2: Make your paper more organized. I cannot get the main point directly. Write simply.
Response 2: According to your instructive suggestions, we have revised the paper and deleted some repeat sections. The modifications were highlighted in red or strikethrough.

Round 2
Reviewer 3 Report
The manuscript entitled, 15N stable isotope labeling PSTs in Alexandrium minutum for application of PSTs as biomarker should be improved before accepted for publication.
1. Page 1, line 15: Be specific, many is ambiguous.
2. Page 2, line 51: Insert references of the related studies.
3. I think you should introduce first what are groups A-E?Why change group A to first generation? or letters to generation number?
4. What is GTX1-4? Please introduce in the Introduction. The whole paper should be correlated for the general readers.
5. Page 8, line 226: Make this statement more scientific.
6. Page 8, line 238: …was not high in what organism?
7. Most sentences are wordy and not scientific. I suggest to make some sentences simple and no repetition.
Author Response
Point 1: Page 1, line 15: Be specific, many is ambiguous.
Response 1: Thank you very much for the comments concerning our manuscript. During this experiment, we cultured A.minitum using 15N-labelled sodium nitrate as nitrogen source continuously about 8 months (15-20 generations). At No.15 generation, we enlarged algae in large scale culture for our another research. Fortunately, we didn’t note down the generation number after No.15 generation and chose one generation of large culture for identifying 15N abundance as No.n. We have noted that “many” is more than 15 in bracket in revised manuscript.
Point 2: Page 2, line 51: Insert references of the related studies.
Response 2: According to your suggestions, we have added some related reference as follows.
Bequette, B.J.; Backwell, F.R.C.; Calder, A.G.; Metcalf, J.A.; Beever, D.E.; Macrae, J.C.; Lobley, G.E. Application of a U- 13C-Labeled Amino Acid Tracer in Lactating Dairy Goats for Simultaneous Measurements of the Flux of Amino Acids in Plasma and the Partition of Amino Acids to the Mammary Gland. Journal of Dairy Science 1997, 80, 2842-2853, doi:10.3168/jds.S0022-0302(97)76249-0. (page 13, lines 422-425)
Cox, J.; Kyle, D.; Radmer, R.; Delente, J. Stable-isotope-labeled biochemicals from microalgae. Trends in Biotechnology 1988, 6, 279-282, doi:10.1016/0167-7799(88)90125-4. (page 13, lines 426-427)
Fernández, F.G.A.; Alias, C.B.; Pérez, J.A.S.; Sevilla, J.M.F.; González, M.J.I.; Grima, E.M. Production of 13C polyunsaturated fatty acids from the microalga Phaeodactylum tricornutum. Journal of Applied Phycology 2003, 15, 229-237, doi:10.1023/a:1023871715805. (page 13, lines 428-430)
Point 3: I think you should introduce first what are groups A-E?Why change group A to first generation? or letters to generation number?
Response 3: To better understand , we have removed the part about what groups A-E are from 5.2. Algal culture to 2.1. Batch culture: 15N/P influence in algal growth and toxicity (page 2, lines 77-81). The letters and numbers are different ways to distinguish batch culture and generation to generation culture. Because Groups A and B grew better than others in batch culture, the two groups were chosen to the experiment of generation to generation culture.
Point 4: What is GTX1-4? Please introduce in the Introduction. The whole paper should be correlated for the general readers.
Response 4: Gonyautoxins 1-4 (GTX1-4) are belonging to mono-sulfated subgroup of PSTs. About this, we made some modification (page 2, lines 71-72) and added a reference Neurotoxic Alkaloids: Saxitoxin and Its Analogs (page 13, lines 438-439) so that general readers can get more detail information about PSTs. Saxitoxin (STX) and its 57 analogs are a broad group of natural neurotoxic alkaloids, commonly known as the paralytic shellfish toxins (PSTs). These analogs differ in side group moieties and thus are commonly grouped according to these variable residues. With more sensitive detection methods, new STX analogs will most likely continue to be identified, with new functional moieties and possibly novel bioactivity.
Point 5: Page 8, line 226: Make this statement more scientific.
Response 5: According to your comment, this sentence has been revised in the revised manuscript (page 8, lines 239-241).
Point 6: Page 8, line 238: …was not high in what organism?
Response 6: 15N-labeling abundance of 15N-PSTs extract was not high than labeling abundance of biomass (whole algae cell) in the reference Cost-effective production of 13C, 15N stable isotope-labelled biomass from phototrophic microalgae for various biotechnological applications. Simply, the 15N-labeling abundance of extract is lower than 15N-labeling abundance of whole cell.
Point 7: Most sentences are wordy and not scientific. I suggest to make some sentences simple and no repetition.
Response 7: According to your suggestions, we have tried our best to revise the paper. The revised manuscript has been read and corrected by a colleague proficient in English.